*Report*

# Altered bioenergetics and mitochondrial dysfunction of monocytes in patients with COVID-19 pneumonia

Lara Gibellini[1,*,†] , Sara De Biasi[1,†] , Annamaria Paolini[1], Rebecca Borella[1], Federica Boraldi[2], Marco Mattioli[1], Domenico Lo Tartaro[1], Lucia Fidanza[1], Alfredo Caro-Maldonado[3], Marianna Meschiari[4], Vittorio Iadisernia[4], Erica Bacca[4], Giovanni Riva[5], Luca Cicchetti[6], Daniela Quaglino[2], Giovanni Guaraldi[4], Stefano Busani[7], Massimo Girardis[7], Cristina Mussini[4] & Andrea Cossarizza[1,8,**]

## Abstract

In patients infected by SARS-CoV-2 who experience an exaggerated inflammation leading to pneumonia, monocytes likely play a major role but have received poor attention. Thus, we analyzed peripheral blood monocytes from patients with COVID-19 pneumonia and found that these cells show signs of altered bioenergetics and mitochondrial dysfunction, had a reduced basal and maximal respiration, reduced spare respiratory capacity, and decreased proton leak. Basal extracellular acidification rate was also diminished, suggesting reduced capability to perform aerobic glycolysis. Although COVID-19 monocytes had a reduced ability to perform oxidative burst, they were still capable of producing TNF and IFN-γ *in vitro*. A significantly high amount of monocytes had depolarized mitochondria and abnormal mitochondrial ultrastructure. A redistribution of monocyte subsets, with a significant expansion of intermediate/pro-inflammatory cells, and high amounts of immature monocytes were found, along with a concomitant compression of classical monocytes, and an increased expression of inhibitory checkpoints like PD-1/PD-L1. High plasma levels of several inflammatory cytokines and chemokines, including GM-CSF, IL-18, CCL2, CXCL10, and osteopontin, finally confirm the importance of monocytes in COVID-19 immunopathogenesis.

**Keywords** COVID-19; inhibitory checkpoints; mitochondria; monocytes; OXPHOS

**Subject Categories** Immunology; Microbiology, Virology & Host Pathogen Interaction

## Introduction

COVID-19 is a complex pathological condition resulting from SARS-CoV-2 infection. This dramatic epidemic started in China in late 2019 and has rapidly spread worldwide (Zhou *et al*, 2020). In Italy, the first patients were observed in Lombardy at the end of February, 2020 (Cossarizza *et al*, 2020a). As of October 9, 2020, the total number of reported cases in Italy was more than 338,000 with more than 36,000 deaths (for constantly updated information, see: https://gisanddata.maps.arcgis.com/apps/opsdashboard/index. html#/bda7594740fd40299423467b48e9ecf6).

Although the precise mechanisms that trigger the pathogenesis of this disease are still poorly known, two different, often concomitant and interlinked, features have been observed, i.e., an exaggerated immune-mediated inflammation and an aberrant coagulation (Cao & Li, 2020). These phenomena can be due to the abnormal production and utilization of a variety of molecules of innate immunity and to the massive activation of cells belonging either to the innate or adaptive branches of immunity (De Biasi *et al*, 2020a; De Biasi *et al*, 2020b). As a result, an auto-aggressive response emerges (Tay *et al*, 2020). In several patients, inflammation is not able to clear the infection as in a normal immune response, and causes severe damages, mainly to the lungs (Merad & Martin, 2020). Thus, one of the main challenges in understanding the immunopathogenesis of

1 Department of Medical and Surgical Sciences for Children and Adults, University of Modena and Reggio Emilia, Modena, Italy
2 Department of Life Sciences, University of Modena and Reggio Emilia, Modena, Italy
3 Agilent Technologies, Madrid, Spain
4 Infectious Diseases Clinics, AOU Policlinico and University of Modena and Reggio Emilia, Modena, Italy
5 Department of Laboratory Medicine and Pathology, AUSL/AOU Policlinico, Modena, Italy
6 Labospace srl, Milan, Italy
7 Department of Anesthesia and Intensive Care, AOU Policlinico and University of Modena and Reggio Emilia, Modena, Italy
8 Institute for Cardiovascular Research, Bologna, Italy
 *Corresponding author. Tel: +39 059 2055726; E-mail: lara.gibellini@unimore.it
 **Corresponding author. Tel: +39 059 2055415; E-mail: andrea.cossarizza@unimore.it
 †These authors contributed equally to this work

COVID-19 is to discover why, despite a potent initial immune response, the infection is not cleared, and what sustains a pathological inflammatory environment.

Inflammation is triggered and regulated by innate cells like monocytes (Ginhoux & Jung, 2014), an heterogeneous population of antigen presenting cells that express MHC class II molecules (i.e., HLA-DR). On the basis of the presence of CD14 and CD16, they can be divided into three different subsets, namely classical, intermediate, and nonclassical monocytes. Classical monocytes express high levels of CD14 but not CD16 (CD14$^{bright}$, CD16$^{-}$), intermediate monocytes express high levels of CD14 and CD16 (CD14$^{bright}$, CD16$^{bright}$), whereas nonclassical monocytes express CD14 and high levels CD16 (CD14$^{dim}$, CD16$^{bright}$) (Ziegler-Heitbrock et al, 2010; Gibellini et al, 2016). Monocyte subsets differ for their ability to present antigens and produce pro-inflammatory cytokines, as well as to express specific homing receptors. Interestingly, they can upregulate the so-called inhibitory receptors belonging to the family of immune checkpoints, such as PD-1 and its ligands (including PD-L1), in several pathological conditions (Zasada et al, 2017; Wang et al, 2017; Tai et al, 2018; Riva & Mehta, 2019).

During the last years, a bioenergetics approach has been used to better understand monocyte regulation and activation (Morton et al, 2019; Yamada et al, 2020). For ATP production, monocytes rely both on oxidative phosphorylation (OXPHOS) and glycolysis (Kramer et al, 2014). However, under conditions that diverge from normal physiology, including inflammation or hypoxia, monocytes activate transcriptional responses that modulate or change cellular metabolism and induce a preferential use of one of the two pathways (Sanmarco et al, 2019). Recent studies have shown that monocyte bioenergetics and inflammatory phenotype can change in response to different infections or pathological stimuli such as hypoxia (Yamada et al, 2020). Other studies demonstrated that microbial stimulation of different Toll-like receptors induced different metabolic programs in monocytes (Lachmandas et al, 2016). For example, after stimulation with whole-pathogen lysates from Escherichia coli, Staphylococcus aureus, and Mycobacterium tuberculosis, and with the TLR2 ligand Pam3CysSK4, monocyte increase oxygen consumption rate (OCR) and glycolysis, which are essential for the activation of host defense mechanisms, such as cytokine production and phagocytosis. Functional and metabolic re-programming have been also observed in monocytes during sepsis and are mediated by hypoxia-inducible factor-1α (HIF1α) (Shalova et al, 2015).

Given the role of inflammation in the immunopathogenesis of COVID-19, we asked whether and how SARS-CoV-2 infection could impact monocyte metabolism and function, thus participating to the immune derangement that does not allow the resolution of the infection and is deleterious for the host. We found that monocytes from COVID-19 patients were metabolically defective, with decreased OXPHOS and glycolysis, and impaired oxidative burst. This was accompanied by an increased expression of inhibitory immune checkpoints, including programmed death-1 (PD-1) and its ligand. A massive alteration of the plasma levels of several cytokines and chemokines involved in inflammation was also observed in COVID-19 patients. Among these cytokines, granulocyte-monocyte colony-stimulating factor (GM-CSF), which drives myelopoiesis, was severely increased. Moreover, a higher percentage of immature monocytes was present in peripheral blood from COVID-19 patients. On the whole, our data suggest that functional and bioenergetics alterations of the monocyte compartment are crucial moments in the pathogenic response to SARS-CoV-2 that is present in patients with COVID-19 pneumonia.

# Results and Discussion

Twenty-eight COVID-19 patients with COVID-19 pneumonia admitted at the University Hospital in Modena (Italy) in March–July 2020 were included in this study. Their main symptoms at admission were fever (79.3% of patients), cough (69%), and dyspnoea (58.6%). Few patients also showed minor symptoms like malaise, myalgia, dysgeusia, diarrhea, and asthenia. The mean concentration of D-dimer, total bilirubin, lactate dehydrogenase (LDH), and C-reactive protein (CRP) were 2,783 ng/ml (range: 390–17,140; normal values: < 250 ng/ml), 0.51 mg/dl (range: < 0.3–1.19; normal values: 0.2–1.2 mg/dl), 576.9 U/l (range: 245–1,022; normal values: 100–190 U/l), and 10.7 mg/dl (range: < 0.2–34.5; normal values: < 1 mg/dl), respectively, indicating the presence of a high of systemic inflammation. Neutrophil-to-lymphocyte ratio, which is associated to disease severity (Liu et al, 2020; Merad & Martin, 2020), was also higher in COVID-19 patients if compared to healthy controls (Table 1).

We first characterized the bioenergetic profile of monocytes from 13 of them to determine whether SARS-CoV-2 infection had an impact on the metabolic capacity of these cells. Monocytes from COVID-19 patients displayed reduced basal and maximal respiration, reduced proton leak, and reduced spare respiratory capacity if compared to those from healthy controls (Fig 1A). The ability of such cells to maintain the maximal respiration was investigated by analyzing the area under the curve (AUC) of the OCR trace from the sixth to the tenth timepoint and was indicated as "maximal respiration kinetic range". Monocytes from COVID-19 patients displayed a reduced capacity to maintain maximal respiration, if compared to controls. A marked reduction of the spare respiratory capacity was also observed in COVID-19 monocytes, suggesting that in these cells the mitochondrial capacity to meet metabolic demands in stressing conditions was compromised. Basal extracellular acidification rate (ECAR) was also reduced in monocytes (Fig 1A). Mitochondrial mass and the percentage of monocytes with depolarized mitochondria were increased in monocytes from COVID-19 patients (Appendix Fig S1A and B). These data demonstrated that both the oxidative and glycolytic metabolism were defective in COVID-19 monocytes, a condition that strongly resembles the "immunometabolic paralysis" observed in monocytes from patients with sepsis (Cheng et al, 2016).

Indeed, real-time monitoring of OCR revealed that when monocytes from patients were challenged with phorbol 12-myristate 13-acetate (PMA)-ionomycin they had a defective respiratory burst if compared to healthy controls (Fig 1A). The respiratory or oxidative burst is the rapid release of reactive oxygen species (ROS) from different cell types, including monocytes, and its quantification is a direct measure of activation and phagocytic function (Vergis et al, 2017). The capacity to perform the oxidative burst in response to a challenge, together with phagocytosis, microbial killing and cytokine production are main functions of monocytes and are influenced

**Table 1. Demographic and clinical characteristics of patients. LDH, lactate dehydrogenase; CRP, C-reactive protein.**

| Variable | COVID-19 patients (n = 28) | Healthy controls (n = 27) |
|---|---|---|
| Age (mean year, range) | 63.0 (37–89) | 58.0 (35–80) |
| Sex (M, %) | 68.9 | 53.5 |
| Symptoms | | |
| Fever (%) | 79.3 | |
| Cough (%) | 69.0 | |
| Dyspnoea (%) | 58.6 | |
| Myalgia (%) | 34.5 | |
| Dysgeusia (%) | 6.9 | |
| Arthralgias (%) | 10.3 | |
| D-dimer (mean, ng/ml) | 2,783.0 | < 250.0 |
| Total bilirubin (mean, mg/dl) | 0.51 | 0.2–1.2 |
| LDH (mean, U/l) | 576.9 | 100–190 |
| CRP (mean, mg/dl) | 10.7 | < 1.0 |
| Blood cell count | | |
| Neutrophils (mean N/μl) | 5,439 | 3,723 |
| Lymphocytes (mean N/μl) | 1,256 | 2,199 |
| Monocytes (mean N/μl) | 430 | 457 |
| Platelets (mean N × $10^3$/μl) | 224 | 303.6 |
| Neutrophil-to-lymphocyte ratio (mean) | 7.2 | 1.28 |

by the metabolic state of the cell (McBride *et al*, 2020). During the oxidative burst, a large amount of oxygen is consumed generating ROS, to kill pathogens. For this reason, to investigate the complete kinetic range of the maximal burst and the immediate response, oxidative burst was measured by calculating the AUC between the tenth and the thirteenth timepoint and by analyzing the oxygen consumption at the eleventh timepoint, respectively. The complete burst, as well as the immediate response, was decreased in monocytes from COVID-19, thus proving that cells from patients were less efficient and dysfunctional (Fig 1A). Recently, the presence of so-called dysfunctional HLA-DR$^{lo}$, CD163$^{hi}$, and HLA-DR$^{lo}$, S100A$^{hi}$, CD14$^+$ monocytes in severe COVID-19 has also been described (Schulte-Schrepping *et al*, 2020). Dysfunctional monocytes were identified on the basis of low HLA-DR expression, but a formal proof of the functional status of monocytes was missing (Schulte-Schrepping *et al*, 2020).

We asked whether, together with alterations in the magnitude of oxidative burst, monocytes from COVID-19 patients also displayed changes in the capability to produce cytokines. For this reason, we quantified the percentage of monocytes able to produce *in vitro* IFN-γ and/or TNF after stimulation with PMA/ ionomycin. The gating strategy is reported in Appendix Fig S2A. We found that, even if metabolically impaired, monocytes from COVID-19 patients were still capable to produce cytokines (Appendix Fig S2B and C). Although not statistically significant, the proportion of unstimulated monocytes producing IFN-γ was slightly decreased in COVID-19 patients if compared to controls,

whereas the proportion of those producing TNF was higher (Appendix Fig S2C). These observations were somewhat unexpected, but likely they could suggest that monocytes from COVID-19 were skewed toward the production of pro-inflammatory cytokines, such as TNF, rather than toward the production of molecules with direct antiviral activity, such as IFN-γ. Similarly, it was reported that TNF/IL-1β-driven inflammatory response was dominant over an IFN-driven response in COVID-19, if compared to severe influenza (Lee *et al*, 2020).

Defects in bioenergetic performance were accompanied by mitochondrial ultrastructural rearrangements, as seen by transmission electron microscopy (Fig 1B). In panels "a" and "b" of Fig 1B, representative electron micrographs of monocytes from healthy subjects and COVID-19 patients are reported. Cells were characterized by an eccentric reniform nucleus (indicated as "N"), chromatin aggregates at the periphery, which were interrupted at the level of nuclear pores. The cytoplasm appeared filled with abundant endoplasmic reticulum, Golgi complex, and small granules. In control cells, mitochondria exhibited well-formed *cristae* (panel "c") and, depending on the functional state, more or less condensed matrix can be observed. Mitochondria of monocytes from COVID-19 patients are shown in panels "d" and "e". These organelles were characterized by heterogeneous size (panels "d"). Moreover, a number of mitochondria displayed intact mitochondrial membranes but high degree of swelling with electron-lucent matrix (panel "e"). Mitochondrial size parameters (area, perimeter, and Feret's diameter) were significantly higher in COVID-19 compared to healthy monocytes (panel "f"). Moreover, the shape of mitochondria was evaluated as aspect ratio, circularity, and roundness. Mitochondria in monocytes of COVID-19 patients had significant changes in aspect ratio and roundness, whereas no difference was observed for circularity (panel "f").

Altogether, our data suggest that monocytes from COVID-19 patients accumulate dysfunctional mitochondria and are metabolically impaired if compared to those from healthy subjects. This could lead to an altered immune response with a possible increased susceptibility to secondary infections. Again, the mitochondrial alterations are reminiscent of those observed in sepsis (Japiassu *et al*, 2011; Arulkumaran *et al*, 2016; Jang *et al*, 2019; McBride *et al*, 2020). Several studies in PBMC from patients with a systemic infection reported significant depolarization of mitochondrial membrane potential, reduced mitochondrial respiration, and severe impairment of mitochondrial electron transport chain complexes I, III, and IV (Japiassu *et al*, 2011; Jang *et al*, 2019; McBride *et al*, 2020). An association between mitochondrial dysfunctions and decreased ATP concentration in skeletal muscles has been correlated with organ failure, thus implying that bioenergetic failure can be a crucial pathological mechanism underlying multiorgan dysfunction (Brealey *et al*, 2002).

To understand whether alterations in the bioenergetics performance were associated with phenotypic alterations, we used polychromatic flow cytometry to analyze the expression of several plasma membrane markers (Cossarizza *et al*, 2019). Monocyte subpopulations in patients and healthy donors were analyzed by two complementary methodologies. In the first, we used the classical approach based upon the two-dimensional recognition of a given cell type, followed by a first gate that was required to identify the population of interest, followed by

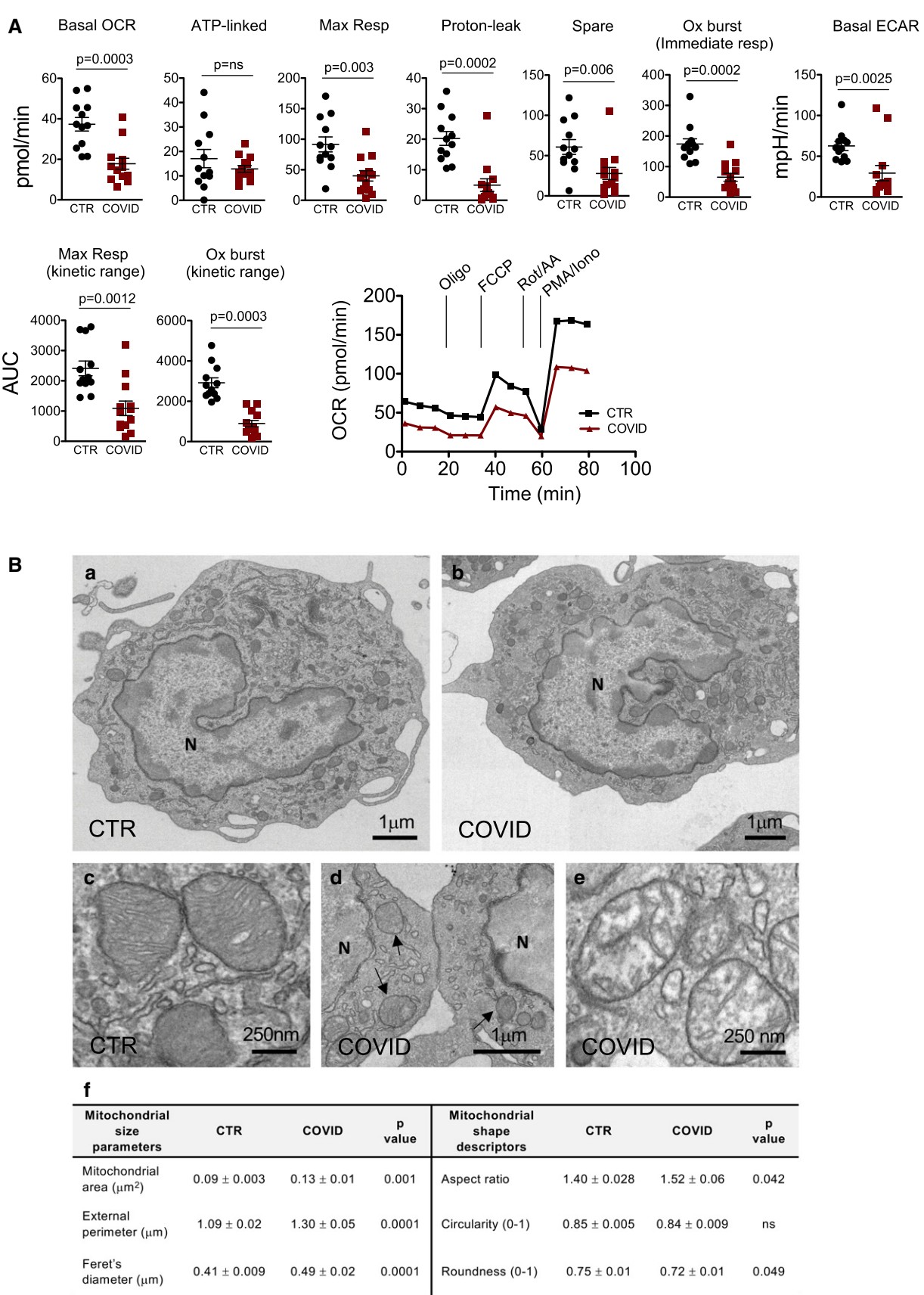

**Figure 1.**

**Figure 1. Monocytes from COVID-19 have impaired bioenergetics and dysfunctional mitochondria.**

A  Measurement of oxygen consumption rate (OCR), extracellular acidification rate (ECAR), and oxidative burst of monocytes from COVID-19 patients (COVID) and healthy controls (CTR). OCR was measured in real time, under basal condition, and in response to mitochondria inhibitors: oligomycin (Oligo, 2 μM), cyanide-4-(trifluoromethoxy)phenylhydrazone (FCCP, 0.5 μM), and antimycin A plus rotenone (Rot/AA, 0.5 μM). Oxidative burst was measured in response to PMA/ionomycin (PMA/Iono, 1 μg/ml). Scatter plots show the quantification of basal respiration (indicated as Basal OCR), ATP-linked respiration (ATP-linked), maximal respiration (Max Resp), proton leak, spare respiratory capacity (Spare), oxidative burst immediate response, and basal ECAR. The maximal respiration kinetic range and the oxidative burst kinetic range are also reported and were obtained by analyzing the area under the curve (AUC) from the sixth to the tenth measurement and from the tenth to the thirteenth measurement, respectively. Data represent individual values, mean and standard error of the mean (CTR, $n = 12$; COVID, $n = 13$). Mann–Whitney test was used for statistical analysis. Exact $P$ values are reported in the figure. Representative traces of OCR of monocytes from COVID-19 patients (COVID) and healthy controls (CTR) are also reported.

B  Representative electron micrographs of mitochondrial ultrastructural changes observed in COVID-19 monocytes. Panels "a" and "c" are representative electron micrographs of monocytes from healthy controls, whereas panels "b", "d", and "e" are representative electron micrographs of monocytes from COVID-19 patients. In panel "f", the quantification of mitochondrial area, external perimeter, Feret's diameter, aspect ratio, circularity, and roundness are reported ($n = 225$ mitochondria for each group). Black arrows in panel "d" indicate mitochondria; N refers to nucleus. Exact $P$ values are reported in panel "f".

sequential gates to identify other markers. Data obtained with this gating strategy were then analyzed by appropriate, non-parametric statistical tests, and represented in the figures as scatter plots with means and standard errors of the mean. In the second, we performed an unsupervised analysis that uses the multidimensional information obtained by FlowSOM meta-clustering coupled to a dimension-reduction method such as the Uniform Manifold Approximation and Projection (UMAP), as described (De Biasi et al, 2020b). Heat maps finally report statistical analysis (see Materials and Methods for details).

Four populations were identified by UMAP: classical monocytes (defined as CD14$^+$, CD16$^-$, CD38$^+$, CCR2$^+$), intermediate monocytes expressing CCR2 (defined as CD14$^+$, CD16$^{++}$, CD38$^+$, CCR2$^+$), intermediate monocytes not expressing CCR2 (defined as CD14$^+$, CD16$^{++}$, CD38$^+$, CCR2$^-$), and nonclassical monocytes (defined as CD14$^-$, CD16$^{++}$, CD38$^-$, CCR2$^-$) (Fig 2A). All subsets of monocytes expressed different levels of C-X-C motif chemokine receptor 3 (CXCR3), PD-1, and PD-L1 (Fig 2B). TIM3 was almost unchanged (Fig 2B). Classical monocytes were 86.7% (median value) of total monocytes in CTR vs. 75.8% in COVID-19 patients. Intermediate monocytes expressing CCR2$^+$ were 2.4% in CTR and 10.9% in COVID-19 patients while those not expressing CCR2 were 2.7% in CTR vs. 4.4% in COVID-19 patients. Nonclassical monocytes were 8.2% in CTR vs. 8.9% in COVID-19 patients (Fig 2B). COVID-19 patients display a different distribution of monocytes subsets if compared to CTR (Fig 2C and D). Interestingly, monocytes expressing high levels of CCR2 are able to migrate to sites of injury where they can differentiate into inflammatory macrophages (Marsh et al, 2017). Furthermore, CCR2$^+$ monocytes, that were increased in COVID-19, have been reported capable to activate innate NK lymphocytes and memory CD8$^+$ T cells which are involved in antiviral immunity (Soudja et al, 2012). Results obtained by unsupervised approach were then confirmed by the classical analysis based on sequential gates (Appendix Fig S3 and S4).

During infection, classical monocytes home to sites of inflammation, recognize and phagocytose pathogens, secrete several pro-inflammatory cytokines, and recruit other immune cells for regulation of the inflammatory response (Ziegler-Heitbrock et al, 2010). Nonclassical monocytes exhibit a distinct motility along the vasculature and are considered patrolling monocytes. Intermediate monocytes mainly exert pro-inflammatory functions. Intermediate monocytes are the main sources of

pro-inflammatory cytokines, including IL-6, TNF, IL-1β, and IL-8, which are increased in plasma from COVID-19 patients, contribute to the cytokine storm (Jose & Manuel, 2020; Cossarizza et al, 2020b), and whose reduction by biological drugs can dramatically decrease mortality (Guaraldi et al, 2020). T-cell immunoglobulin and mucin-domain containing-3 (TIM-3), PD-1, and PD-L1 are three molecules that regulate cell-mediated immunity and are expressed by different immune cells, including lymphocytes and monocytes (Anderson et al, 2007; Keir et al, 2008; Xia et al, 2018). Recent findings suggest that the PD-1/PD-L1 pathway plays an important role in the pathogenesis of sepsis, with increased expression of PD-1 or PD-L1 on monocytes being associated with increased mortality in septic patients (Zhang et al, 2010; Zasada et al, 2017; Tai et al, 2018).

TIM-3 is expressed on monocytes in different clinical and pathological settings, and it can synergize with Toll-like receptors (Anderson et al, 2007; Riva & Mehta, 2019). The precise role of TIM-3 in monocyte biology is controversial and still scarcely understood. However, in general, expression of inhibitory immune checkpoints, like PD-1, PD-L1, and TIM-3, has been linked to a more suppressive phenotype and worse prognosis in sepsis, infections, and even cancer (Zasada et al, 2017; Wang et al, 2017; Tai et al, 2018; Riva & Mehta, 2019). We found that in COVID-19 patients, a higher percentage of PD-1$^+$ and PD-L1$^+$ cells were present among total monocytes, as well as within different subsets (Fig 2E, and Appendix Fig S5A and B). We could also measure the amount of molecules per cell by the quantification of median fluorescence intensity (MFI), and we found that the expression of PD-L1 on monocytes from COVID-19 patients was also increased (Fig 2E and Appendix Fig S5C). It is interesting to note that despite the heterogeneity characterizing circulating monocytes, inhibitory checkpoints, in particular PD-1 and PD-L1, were significantly over-expressed in all the three subsets of monocytes, while previous data revealed that PD-L1 was mainly expressed on nonclassical monocytes in peripheral blood (Bianchini et al, 2019). PD-1 and TIM-3 and their ligands, PD-L1 and galectin-9, are inhibitory pathways that regulate the balance between protective immunity and host immune-mediated damage. In a recent study, we found a dramatic increase of galectin-9 in the plasma of patients with COVID-19 pneumonia (De Biasi et al, 2020b). We also showed that soluble PD-L1 is present at thigh

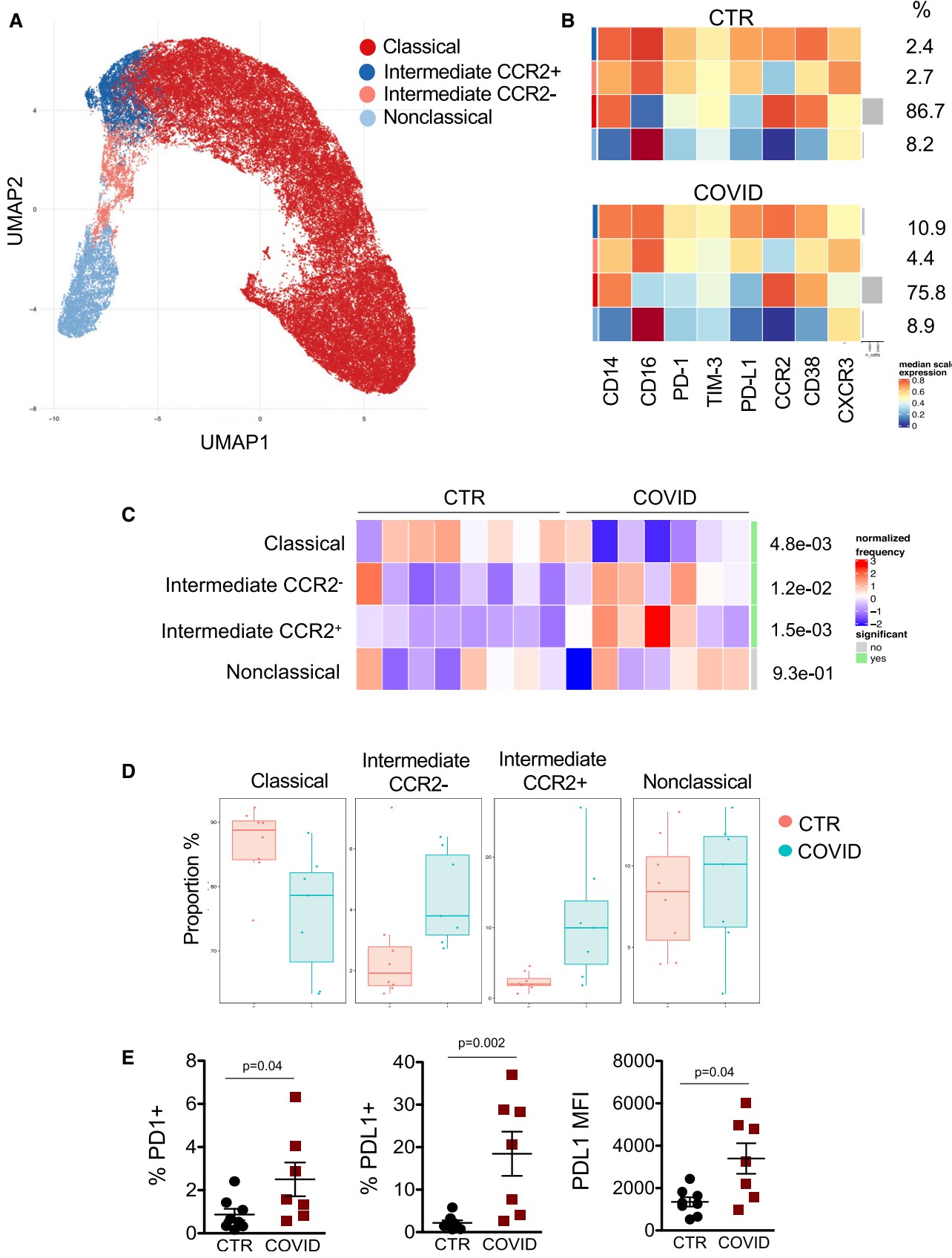

**Figure 2.**

**Figure 2.  Unsupervised analysis of monocyte subsets in CTR and COVID-19 patients.**

A   Uniform Manifold Approximation and Projection (UMAP) representation of monocyte subpopulations.
B   Heatmaps representing different monocyte clusters identified by FlowSOM, with percentages in healthy donors (CTR) and COVID-19 patients. The colors in the heat map represent the median of the arcsinh, 0–1 transformed marker expression calculated over cells from all the samples, varying from blue for lower expression to red for higher expression. Each cluster has a unique color assigned (bar on the left which colors correspond to those used in panel A).
C   Differential analysis between healthy donors (CTR, *n* = 8) and COVID-19 (COVID, *n* = 7). The heatmap represents arcsine-square-root transformed cell frequencies that were subsequently normalized per cluster (rows) to mean of zero and standard deviation of one. The color of the heat varies from blue, indicating relative under-representation, to red indicating relative over-representation. Bar and numbers at the right indicate significant differentially abundant clusters (green) and adjusted *P* values (Differential analysis performed by diffcyt package present in the Catalyst package). The statistical analyses were done using GLMM (according to the actual guidelines), and all *P* values were Bonferroni corrected. Clusters were sorted according to the adjusted *P* values, so that the cluster at the top shows the most significant abundance changes between the two conditions.
D   Boxes and whiskers plots representing maximum, minimum, median value, 25 and 75 percentile of different subpopulations of monocytes in healthy donors (CTR, *n* = 8) and COVID-19 patients (COVID, *n* = 7).
E   Scatter plots showing the frequency of PD-1$^+$ cells, PD-L1$^+$ cells among total monocytes in healthy controls (CTR, *n* = 8) and COVID-19 patients (COVID, *n* = 7). A scatter plot reporting the median fluorescence intensity (MFI) of PD-L1 in total monocytes is also shown. Data represent individual values, mean ± standard error of the mean. Mann–Whitney test was used for statistical analysis. Exact *P* values are reported in the figure.

concentration in plasma from COVID-19 patients, if compared to healthy controls (Fig 3A). Thus, even if in this study we found a similar expression of TIM-3 on monocytes from healthy controls and COVID-19 patients, we can hypothesize that also in these patients soluble PD-L1 and galactin-9 can engage their receptors on monocytes, as already described in other settings, including sepsis and alcoholic liver disease, and promote immune exhaustion and paralysis (Markwick *et al*, 2015).

The upregulation of PD-1 and PD-L1 on monocytes from COVID-19 raises important clinical questions for the potential intersection between COVID-19 and cancer immunotherapy, being immune checkpoint inhibitors (ICI) the pillar of cancer therapy in several tumors (Maio *et al*, 2020). On the one hand, in the presence of the cytokine storm, ICI therapy could affect the monocytic compartment, further unbalancing the immunologic response. This, in turn, could exacerbate inflammation and therefore worsen the clinical course of COVID-19 disease. To this regard, it has been found that anti-PD-L1 therapy led to dominant gene expression changes in CD14$^+$ monocytes (Bar *et al*, 2020). On the other hand, considering the T-cell compartment, ICI therapy could mitigate the early phase of COVID-19 disease by contributing to viral clearance through the reactivation of potentially exhausted, PD-1$^+$ antigen-specific T cells (Maio *et al*, 2020). These considerations imply that scientific evidences are now needed to provide mechanistic insights on the possible relationship between ICI and COVID-19 infection and to clarify whether ICI could be used in cancer patients with concomitant COVID-19.

Then, to further investigate the role of circulating monocytes and their environment in COVID-19-related inflammation, we measured plasma concentrations of IL-6, TNF, IFN-γ, IL-18, C-C motif chemokine ligand 2 (CCL2), CCL11, GM-CSF, osteopontin (OPN), and C-X-C motif chemokine ligand 10 (CXCL10), which are involved in monocyte regulation and migration (Fig 3A). We confirmed that plasma concentrations of IL-6, TNF, and CCL2 were higher in COVID-19 patients, as previously observed by several groups, including ours (De Biasi *et al*, 2020b). The presence of high levels of IL-6, TNF, IFN-γ, IL-18 in plasma from COVID-19 patients clearly revealed that an abnormal inflammatory response was present in most individuals. The concentration

of inflammatory chemokines, including CCL2 (ligand of CCR2), CCL11, CXCL10 (ligand of CXCR3), and OPN, was indeed significantly higher if compared to healthy controls. GM-CSF was also increased in COVID-19 patients if compared to CTR (Fig 3A), suggesting the presence of emergency myelopoiesis in severe COVID-19 patients (Silvin *et al*, 2020). Emergency myelopoiesis is a well-known phenomenon, characterized by the mobilization of immature myeloid cells to restore functional immune cells. Thus, we investigated the presence of immature monocytes in peripheral blood. The gating strategy used to identify circulating immature monocytes is reported in Appendix Fig S6. Monocytes were first identified according to physical parameters, and then doublets and dead cells were removed. HLA-DR$^+$ cells were selected and, among these, CD14$^+$, CD13$^+$, CD64$^+$ events were identified as mature monocytes, while CD14$^-$, CD13$^-$, CD64$^-$ were identified as immature monocytes. As expected, immature monocytes were significantly increased in peripheral blood from COVID-10 patients (24.01% ± 8.10 vs. 6.51% ± 0.85) (Fig 3B).

We are aware that our study has some limitations, such as the relatively low number of patients and the number of analyses that we could perform. Furthermore, it would have been important to test the *in vitro* efficacy of different drugs able to influence cellular metabolism. Studies are however in course to correlate the aforementioned parameters as well as the clinical outcomes to the functional exhaustion of the cells.

In conclusion, to the best of our knowledge, this is the first demonstration that in COVID-19 patients the monocyte compartment is severely impaired from the functional and bioenergetics point of view. The impairment was phenotypic, metabolic, and functional, in a process similar to the "immunometabolic paralysis" that accompanies immune changes during sepsis (Rubio *et al*, 2019). We demonstrated that monocytes had broad defects in metabolic pathways, not only failing to increase glycolysis but also exhibiting reduced oxygen consumption rate, together with important mitochondrial dysfunction. Immature monocytes are also released in the periphery. From the phenotypic point of view, the upregulation of inhibitory checkpoints, including PD-1 and PD-L1, exists that likely renders monocytes potential targets of immunotherapy, already available and successfully used in oncology.

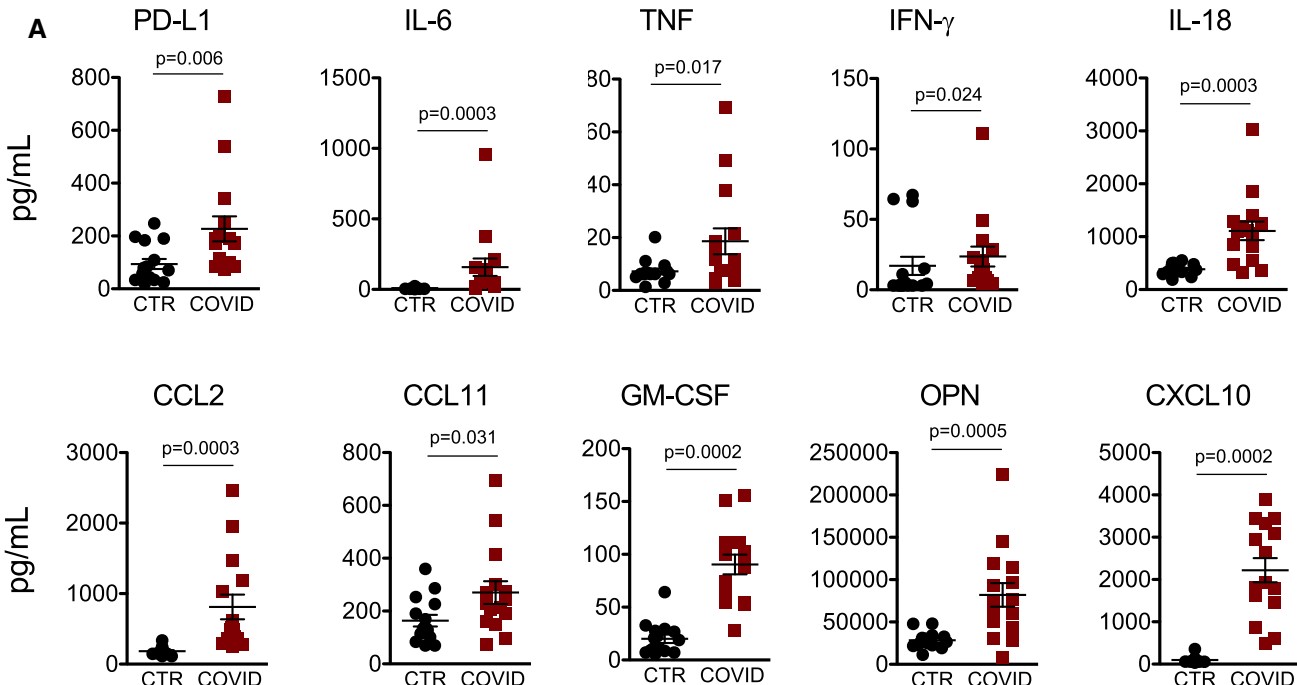

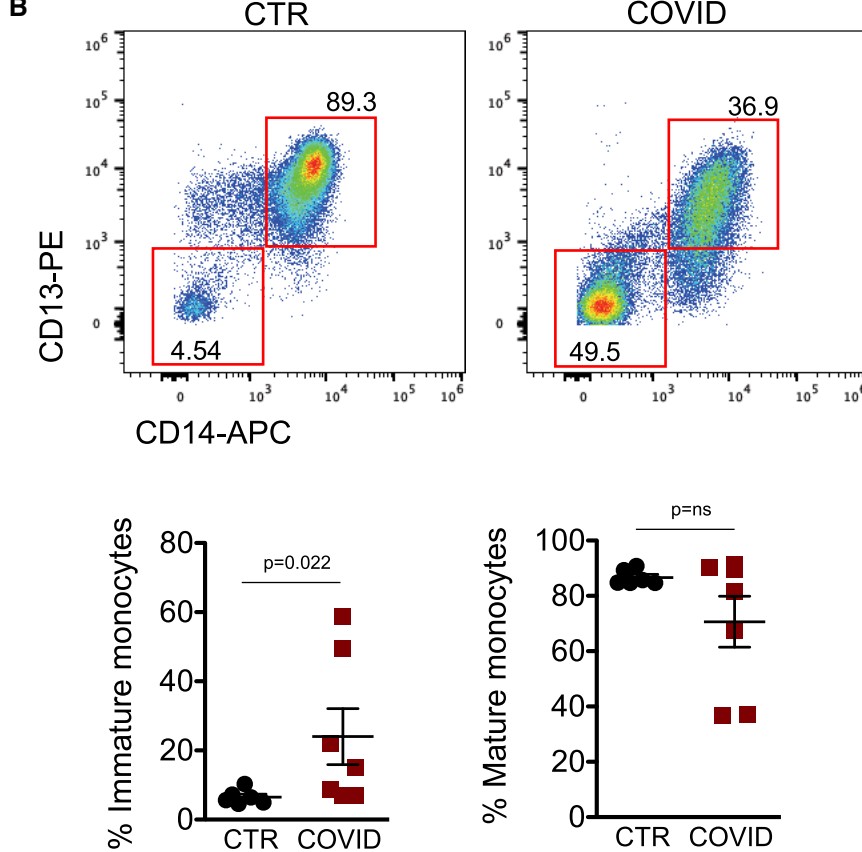

**Figure 3.**

◀

**Figure 3. Plasma levels of cytokines and chemokines involved in monocytes' regulation and appearance of immature monocytes in COVID-19 patients' peripheral blood.**

A  Quantification of cytokines and other mediators in plasma obtained from COVID-19 patients (COVID, $n = 15$) and healthy controls (CTR, $n = 15$). Data represent individual values, mean ± standard error of the mean. Mann–Whitney test was used for statistical analysis. Exact $P$ values are reported in the figure.

B  Representative dot plots showing percentages of immature and mature circulating monocytes in CTR and COVID-19 patients. Scatter plots showing the quantification of immature and mature monocytes in healthy controls (CTR, $n = 6$) and COVID-19 patients (COVID, $n = 7$) are reported. Mann–Whitney test was used for statistical analysis. Exact $P$ values are reported in the figure.

**Table 2. List of monoclonal antibodies and fluorochromes for the cytofluorimetric analysis of monocytes.**

| Specificity | Dye | Clone | Manufacturer | Cat. | Lot | Dil. (µl) |
|---|---|---|---|---|---|---|
| PromoFluor-840 | Maleimide | N/A | PromoKine | PK-PF840-3-01 | 429P0-17 | 0.3 |
| CD14 | APC | 63D3 | BioLegend | 367118 | B262993 | 0.6 |
| CD16 | AF488 | 3G8 | BioLegend | 302019 | B216866 | 0.6 |
| HLA-DR | PE-CY7 | L243 | BioLegend | 307616 | B260010 | 0.3 |
| PD-1 | BV421 | EH12.2H7 | BioLegend | 329920 | B217577 | 2.5 |
| TIM-3 | BV785 | F38-2E2 | BioLegend | 345032 | B241706 | 5.0 |
| PD-L1 | PE | MIH1 | Thermo Fisher | 12-5983-41 | 1998319 | 5.0 |
| CD11b | PE-CY5 | ICRF44 | BioLegend | 301308 | B284850 | 2.5 |
| CCR2 | BV605 | K036C2 | BioLegend | 357214 | B253733 | 0.6 |
| CD15 | PE-CY5 | W6D3 | BioLegend | 323014 | B287884 | 2.5 |
| CD38 | BUV496 | HIT2 | Beckton Dickinson | 564658 | 9067622 | 1.25 |
| CXCR3 | BUV395 | 1C6/CXCR3 | Beckton Dickinson | 565223 | 7283777 | 2.5 |
| TNFα | BV605 | MAb11 | BioLegend | 502936 | B282176 | 3.75 |
| IFNγ | FITC | B27 | BioLegend | 506504 | B286029 | 2.5 |
| CD64 | FITC | 10.1 | Thermo Fisher | CD6401 | 1707112A | 5.0 |
| CD13 | PE | L138 | BD | 347406 | 0086623 | 5.0 |
| LIVE/DEAD | AQUA | | Thermo Fisher | L34957A | 2145055 | 1.25 |

Cat, category number; Dil, dilution; Lot, lot number.

# Materials and Methods

### Human samples

Twenty-eight patients with COVID-19 pneumonia admitted at the University Hospital in Modena (Italy) in March–July 2020 were included in this study. They had a median age of 63 years (range 37–89), 68.9% were males. Among them, a sub-cohort of thirteen patients was used to perform bioenergetics, ultrastructural analysis of mitochondria, quantification of monocyte subsets, immature monocytes, and *in vitro* production of cytokines. A sub-cohort of fifteen patients was used to analyze plasma levels of the indicated cytokines and chemokines. Besides, 27 healthy individuals were recruited from the University Hospital personnel during the same period and served as normal controls (median age 58 years; range 35–80 years; 53.5% were males). Among them, a sub-cohort of 12 healthy donors was used to perform bioenergetics, ultrastructural analysis of mitochondria, quantification of monocyte subsets, immature monocytes, and *in vitro* production of cytokines. A sub-cohort of fifteen healthy controls was used to analyze plasma levels of the indicated cytokines and chemokines. We recorded demographic data, medical history, symptoms, signs, temperature, and main laboratory findings from each patient (Table 1). The study protocol was approved by the local Ethical (Area Vasta Emilia Romagna, protocol number 177/2020, March 10, 2020). Written informed consent was obtained from each subject, and the experiments were conformed to the principles set out in the WMA Declaration of Helsinki and the Department of Health and Human Services Belmont Report.

### Blood collection and PBMC isolation

Up to 20 ml of blood was collected from patients and healthy controls in vacuette containing ethylenediamine-tetraacetic acid (EDTA). Blood was immediately processed. Isolation of peripheral blood mononuclear cells (PBMC) was performed using ficoll-hypaque according to standard procedures (De Biasi *et al*, 2019). Plasma was collected, centrifuged, and stored at −80°C.

### Bioenergetics studies

Monocytes were magnetically purified from PBMCs by positive selection using CD14 microbeads (Miltenyi Biotec, Bergisch Gladbach, Germany), according to manufacturer's instructions. Oxygen consumption rate (OCR), extracellular acidification rate (ECAR),

and oxidative burst were quantified by using the Seahorse XFe96 Analyser (Agilent Technologies). In particular, 400,000 monocytes/well were plated in triplicate in XFe96 cell culture microplates (Agilent Technologies) precoated with poly-L-lysine. RPMI medium was replaced with 180 μl of DMEM XF base medium, pH 7.4 (Agilent Technologies) supplemented with 2 mM glutamine, 10 mM glucose, 1 mM pyruvate (All from Agilent Technologies). Plates were kept 30 min at 37°C and loaded into the Seahorse XFe96 Analyser. OCR was quantified at the beginning of the assay and after the sequential injection of 2 μM oligomycin, 0.5 μM Carbonyl cyanide-4-(trifluoromethoxy)phenylhydrazone (FCCP) and 0.5 μM of rotenone plus antimycin A (all from reagents Mito Stress Test kit, Agilent Technologies). Respiratory parameters were obtained as indicated: basal respiration as baseline OCR; proton leak by subtracting OCR after oligomycin injection to basal OCR; ATP-linked respiration by subtracting the proton leak to the basal OCR; spare respiratory capacity as the difference between the maximal respiration and the basal respiration; maximal OCR by calculating the difference of antimycin plus rotenone rate from FCCP rate (Gibellini *et al*, 2018; De Biasi *et al*, 2019). The capacity of cells to maintain the maximal respiration was determined by assaying the area under the curve (AUC) from the sixth to tenth measurement. Oxidative burst was quantified after the injection of phorbol 12-myristate 13-acetate (PMA)/ionomycin (Sigma Aldrich, St. Louis, Missouri, USA) and was determined by assaying the AUC from the tenth to thirteenth measurement. Basal ECAR was determined by analyzing the pH at the baseline.

### Flow cytometry

To identify monocytes, the following procedure was performed. Freshly isolated PBMC were stained with viability marker Promo-Kine 840 (PromoCell, Heidelberg, Germany); then, samples were washed with FACS buffer (PBS, 2% FBS) and incubated with a mix of pre-titrated directly conjugated mAbs. The mix included the following: anti-CD14-APC, anti-CD16-AF488, anti-HLA-DR-PE-Cy7, anti-PD1-BV421, anti-TIM-3-BV785, anti-CD15-PE-Cy5, anti-CD11-PE-Cy5, anti-CCR2-BV605 (BioLegend, San Diego, CA), anti-CD38-BUV496, anti-CXCR3-BUV395 (Becton Dickinson, San José, CA), anti-PD-L1-PE (Thermo Fisher, Eugene, OR). Samples were acquired by using CytoFLEX LX (Beckman Coulter, Hialeah, FL). To identify circulating mature and immature monocytes, thawed PBMC were stained with LIVE-DEAD Aqua, anti-HLA-DR-PE-Cy7, anti-CD14-APC, anti-CD13-PE, anti-CD64-FITC. Table 2 reports mAbs clones, catalog numbers, type of fluorochrome used, and mAbs dilutions. Mitochondrial mass was analyzed by staining cells with Mito-Tracker green (MT Green, Thermo Fisher) (De Biasi *et al*, 2019). Mitochondrial membrane potential was analyzed by staining cells with 1,1′,3,3′-tetraethyl-5,5′,6,6′-tetrachloroimidacarbocyanine iodide (JC-1, Thermo Fisher) (Cossarizza *et al*, 2019). Cells were acquired using Attune NxT Acoustic flow Cytometer.

### *In vitro* stimulation of monocytes and intracellular cytokine staining

Thawed PBMC were washed twice with complete culture medium (RPMI 1640 supplemented with 10% fetal bovine serum and 1%

each of L-glutamine, sodium pyruvate, nonessential amino acids, antibiotics, 0.1 M HEPES, 55 μM β-mercaptoethanol) plus 0.02 mg/ml DNAse. PBMCs were stimulated for 4 h at 37°C in a 5% CO2 atmosphere with PMA (100 ng/ml) and Ionomycin (1 μg/ml) in complete culture medium. For each sample, at least 2 million cells were left unstimulated as negative control, and 2 million cells were stimulated. All samples were incubated with a protein transport inhibitor containing brefeldin A (Golgi Plug, Becton Dickinson). After stimulation, cells were stained with LIVE-DEAD Aqua (Thermo Fisher Scientific) and surface mAbs recognizing HLA-DR-PE-Cy7, CD14-APC, and CD16-BV421 (BioLegend, San Diego, CA, USA). Cells were washed with stain buffer, fixed, and permeabilized with the cytofix/cytoperm buffer set (Becton Dickinson) for cytokine detection. Then, cells were stained with previously titrated mAbs recognizing IFN-γ-FITC and TNF-BV605 (all mAbs from BioLegend). Samples were acquired on Attune NxT acoustic cytometer (Thermo Fisher).

### Transmission electron microscopy

Monocytes were isolated by using the Pan Monocyte Isolation kit (Miltenyi Biotech) and were immediately fixed with 2.5% glutaraldehyde (Electron Microscopy Sciences) in 0.1 M cacodylate buffer (pH 7.4) for 4 h at 4°C, washed in 0.1 M cacodylate buffer, and post-fixed with 1% $OsO_4$ in cacodylate buffer for 1 h at 4°C. The cells were dehydrated by routine procedures (Martín-Montañez *et al*, 2019) and embedded in Epon 812 (Electron Microscopy Sciences). Ultrathin sections (60 nm) were obtained with diamond knife, mounted on copper grids, and stained with a mix of lanthanum salts, samarium triacetate, and gadolinium triacetate (TAAB Laboratories Equipment Ltd) and with Reynolds' lead citrate. The samples were examined with a FEI NOVA NanoSEM 450, and images were obtained using the STEM mode using Solid State Detector with voltage at 30 kV.

Mitochondrial size parameters and mitochondrial shape descriptors were measured on 225 mitochondria (from CTR, $n = 3$; COVID, $n = 3$) for each experimental group by using ImageJ software (version 2.0.0). In particular, we considered mitochondrial area, external perimeter, Feret's diameter (the longest distance between any two points of the mitochondrial external perimeter), aspect ratio (ratio between major and the minor axis of each mitochondrion), and two indexes of sphericity: circularity $[4\pi\cdot(area)/(perimeter^2)]$ and roundness $[4\cdot(area)/\pi\cdot(major\ axis^2)]$ (Errea *et al*, 2015). Data were expressed as mean ± SEM.

### Quantification of cytokine plasma levels

The plasma levels of 10 molecules were quantified using a Luminex platform (Human Cytokine Discovery, R&D System, Minneapolis, MN) for the simultaneous detection of the following molecules: IL-6, TNF, IFN-γ, GM-CSF, PD-L1, IL-18, OPN, CCL2, CCL11, CXCL10, according to the manufacturer's instruction. Data in the scatter plots represent the mean of two technical replicates.

### Statistical analysis

Flow Cytometry Standard (FCS) 3.0 files were imported into FlowJo software version X (Becton Dickinson, San Josè, CA) and analyzed

**The paper explained**

**Problem**

In COVID-19, an exaggerated inflammation leading to the damage of different tissues, including lung, is observed in most patients. Inflammation is initiated and controlled by monocytes, but at present, their role in the immunopathogenesis of this disease is still unexplored.

**Results**

We studied monocytes from COVID-19 patients and found that were metabolically impaired, with decreased respiration and glycolysis. They were less capable to perform oxidative burst if compared to monocytes from healthy controls, but they were still able to produce cytokines. The mitochondrial ultrastructure was profoundly altered. We also identified relevant changes in the distribution of monocyte subsets, with an expansion of intermediate cells, which have pro-inflammatory function, and a reduction of nonclassical monocytes. Monocytes from every subset expressed high levels of inhibitory receptors, including PD-1 and PD-L1. A higher proportion of immature monocytes were observed. A massive alteration of plasma levels of molecules crucial for monocyte regulation was present in COVID-19 patients.

**Impact**

Our data suggest that infection with SARS-CoV-2 could heavily affect the monocytic compartment of innate immunity.

by standard gating to eliminate aggregates and dead cells, and to identify $CD15^+$, $CD11b^+$, $HLA-DR^{+,+/-}$, $CD14^+$ monocytes. Then, data from 3000 monocytes per sample were exported for further analysis in R, by following a script that makes use of Bioconductor libraries and R statistical packages (CATALYST 1.12.2). The script is available at: https://github.com/HelenaLC/CATALYST) (Nowicka *et al*, 2019). The platform FlowSOM was used to perform the meta-clustering ($K = 10$). Data were subsequently displayed using Uniform Manifold Approximation and Projection (UMAP) (Becht *et al*, 2019). Data are represented as individual values, means, and standard errors of the mean. Variables were compared using Mann–Whitney test, and statistical analyses were performed using Prism 8.0 (GraphPad software Inc, La Jolla, USA). There was no randomization, since this is not a clinical trial but a case–control study that compares patients with COVID pneumonia vs. age- and sex-matched healthy donors.

## Data availability

The cytofluorimetric source data underlying Figs 2 and 3, and Appendix Figs S3–S6 are provided as a Source Data file. Original.fcs files are deposited at the https://flowrepository.org/ in the following folder: FR-FCM-Z328 (https://flowrepository.org/id/FR-FCM-Z328).

Expanded View for this article is available online.

## Acknowledgements

Sara De Biasi and Lara Gibellini are Marylou Ingram Scholar of the International Society for Advancement of Cytometry (ISAC) for the period 2015-2020 and 2020-2025, respectively. We gratefully acknowledge Drs. Paola Paglia (Thermo Fisher Scientific, Monza, Italy), Leonardo Beretta (Beckman Coulter, Milan, Italy), Emma Di Capua (Agilent Technologies, Italy) for their continuous and enthusiastic support, especially in the days of lock down, and for precious suggestions. The study has been partially funded by unrestricted donations from: Glem Gas SpA (San Cesario, MO, Italy), Sanfelice 1893 Banca Popolare (San Felice sul Panaro, Modena, Italy) and Rotary Club Distretto 2072 (Clubs in Modena, Modena L.A. Muratori, Carpi, Sassuolo, Castelvetro di Modena), C.O.F.I.M. SPA & Gianni Gibellini, Franco Appari, Assicuratrice Milanese, Andrea Lucchi, Federica Vagnarelli, Biogas Europa Service & Massimo Faccia, Pierangelo Bertoli Fans Club and Alberto Bertoli, Maria Santoro, Valentina Spezzani, and finally by Gruppo BPER (Modena, Italy). They are all strongly acknowledged for their generous support to our research. We also gratefully thank all the many people who have and are helping us with their precious donations. This study was partially supported by Ministero della Salute, Bando Ricerca COVID-19 (2020-2021) to AC, grant number: COVID-2020-12371808. The authors acknowledge the "Fondazione Cassa di Risparmio di Modena" for funding FEI NOVA NanoSEM 450 at the Centro Interdipartimentale Grandi Strumenti (CIGS) of the University of Modena and Reggio Emilia. Finally, a special thanks to the patients who donated blood to participate to this study.

## Author contributions

Conceptualization: LG, SDB, and AC; Methodology: SDB, AP, RB, MMa, AC-M, LC, GR; Computational analysis: DLT; Investigation: LG, DLT, LF, FB, DQ; Supervision and patients' recruitment: MMe, GG, CM, MG, VI, EB, SB, DQ; Funding Acquisition: MG, AC; Writing—Review & Editing: LG, SDB, DQ, AC.

## Conflict of interest

Alfredo Caro-Maldonado is a Seahorse XF Consumables Product Specialist; Luca Cicchetti runs Labospace, operating a service for detection of cytokines. The other authors declare no conflict of interests.

## Additional information

i   The International Society for Advancement of Cytometry—ISAC provides continuous education in the field of different types of cytometric analysis (see: https://isac-net.org).

ii  The Second Edition of the "Guidelines for the use of flow cytometry and cell sorting in immunological studies", are freely available at https://onlinelibrary.wiley.com/doi/full/10.1002/eji.201970107.

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
