## [Review Process File · EMBO Molecular Medicine]

Appendix

Altered bioenergetics and mitochondrial dysfunction
of monocytes in patients with COVID-19 pneumonia

Gibellini et al.

Table of content

Appendix Figure S1

Mitochondrial mass and mitochondrial membrane potential in monocytes from healthy controls and COVID-19 patients

Appendix Figure S2

Cytokine-producing monocytes in healthy controls and COVID-19 patients after *in vitro* stimulation with PMA/Ionomycin

Appendix Figure S3

Gating strategy for the identification of classical, intermediate, nonclassical monocytes expressing programmed death 1 (PD-1), programmed death ligand-1 (PD-L1), T cell immunoglobulin- and mucin-domain-containing molecule-3 (TIM-3), CD38 and C-C motif chemokine receptor 2 (CCR2)

Appendix Figure S4

Distribution of the different subsets of monocytes, according to the expression of CD14 and CD16.

Appendix Figure S5

Expression of programmed death 1 (PD-1) and programmed death ligand-1 (PD-L1) in the different subsets of monocytes

Appendix Figure S6

Gating strategy for the identification of immature monocytes

Appendix Figure S1.

A. Representative histograms showing Mitotracker Green (Mt Green) fluorescence in monocytes from COVID-19 patients (indicated as COVID) and healthy controls (CTR). The scatter plot on the right shows the quantification of Median Fluorescence Intensity (MFI) of Mt Green, as indicator of mitochondrial mass. CTR, $n=3$; COVID, $n=4$. Mann-Whitney test was used for statistical analysis. B. Representative dot plots showing JC-1 monomers and aggregates in monocytes from COVID-19 patients (indicated as COVID) and healthy controls (CTR). The scatter plot on the right shows the quantification of cells with depolarized mitochondria. CTR, $n=6$; COVID, $n=5$. Mann-Whitney test was used for statistical analysis.

Appendix Figure S2.

A. Gating strategy for the identification of cytokine-producing monocytes after *in vitro* stimulation with PMA/Ionomycin (PMA/Iono). PBMC were identified according to physical parameters; doublets and dead cells were removed. Monocytes were identified according to the expression of HLA-DR and CD14. The percentages of monocyte producing IFN- γ and TNF were evaluated before (indicated as unstimulated), and after stimulation. B. Representative dot plots of monocytes producing cytokines after *in vitro* stimulation with PMA/Ionomycin in CTR and COVID patients. C. Scatter plots reporting the quantification of monocytes producing IFN- γ , TNF or both cytokines in COVID-19 patients (COVID, n=8) and healthy controls (CTR, n=11), without any stimulus and after stimulation with PMA/ionomycin. Mann-Whitney test for used for statistical analysis, p=ns.

Appendix Figure S3.

The gating strategy for the identification of classical, intermediate, nonclassical monocytes expressing programmed death 1 (PD-1), programmed death ligand-1 (PD-L1) and T cell immunoglobulin- and mucin-domain-containing molecule-3 (TIM-3) is reported. Peripheral blood mononuclear cell (PBMC) were identified according to physical parameters, single cells were selected according to FSC-H and FSC-W gates, and the eventual turbulences of the flux present during acquisition were removed from the analysis, along with dead cells. Then, cells positive for HLA-DR, CD15 and CD11b were selected and monocytes expressing CD14 were identified. On the basis of the expression of CD14 and CD16, classical monocytes (defined as CD14⁺⁺,CD16⁻), intermediate monocytes (defined as CD14⁺⁺,CD16⁺⁺), and nonclassical monocytes were identified (defined as CD14⁻,CD16⁺⁺). The expression of PD-1, PD-L1, TIM3, C-X-C motif chemokine receptor 3 (CXCR3), C-C motif chemokine receptor 2 (CCR2), and CD38 was evaluated in total monocytes and in the different subsets.

Appendix Figure S4.

Representative dot plots of the different subsets of monocytes, according to the expression of CD14 and CD16. Gates indicate classical (CD14⁺⁺,CD16⁻), intermediate (CD14⁺⁺,CD16⁺⁺), and non-classical (CD14⁻,CD16⁺⁺) monocytes from one healthy control (CTR) and one COVID-19 patient (COVID). Scatter plots report the frequency of classical, intermediate and non-classical monocytes (CTR, n=8; COVID, n=7). Data represent individual values, mean and standard error of the mean. Mann-Whitney test was used for statistical analysis. The intermediate monocytes subset was expanded in COVID-19 vs controls (12.5±3.1% vs 4.5±1.04%, p<0.05), whereas the frequency of classical monocytes only tended to decrease (80.7±1.8% vs 71.7±3.2%, p= not significant). No obvious differences were found concerning the expression of CCR2, CXCR3 and CD38.

Appendix Figure S5.

A. Representative dot plots reporting the expression of PD-1 in classical monocytes from one healthy control (CTR) and one COVID-19 patient (COVID). Scatter plots report the frequency of PD-1⁺ cells among classical, intermediate and nonclassical monocytes. CTR, n=8; COVID, n=7. Mann-Whitney test was used for statistical analysis. B. Representative dot plots reporting the expression of PD-L1 in intermediate monocytes from one healthy control (CTR) and one COVID-19 patient (COVID). Scatter plots report the frequency of PD-L1⁺ cells among classical, intermediate and nonclassical monocytes. CTR, n=8; COVID, n=7. Mann-Whitney test was used for statistical analysis. C. Representative histogram reporting the expression of PD-L1 in monocytes from one healthy control (in grey) and one patient (in red). Scatter plots indicate the quantification of PD-L1 median fluorescence intensity (MFI) in classical, intermediate and nonclassical monocytes. CTR, n=8; COVID, n=7. Mann-Whitney test was used for statistical analysis.

Appendix Figure S6.

The gating strategy for the identification of immature monocytes is reported. Monocytes were identified according to physical parameters, doublets and dead cells were removed. HLA-DR⁺ events were selected and among these, CD14⁺,CD13⁺, CD64⁺ events were identified as mature monocytes and those CD14⁻,CD13⁻,CD64⁻ were identified as immature monocytes.